# Dissemination of COVID-19 in inland cities of Northeastern Brazil

**Sanderson José Costa de Assis**[1☯]*, **Johnnatas Mikael Lopes**[2☯], **Bartolomeu Fagundes de Lima Filho**[3☯], **Geronimo José Bouzas Sanchis**[1☯], **Thais Sousa Rodrigues Guedes**[4☯], **Rafael Limeira Cavalcanti**[5☯], **Diego Neves Araujo**[6☯], **Antonio José Sarmento da Nóbrega**[3☯], **Marcello Barbosa Otoni Gonçalves Guedes**[3☯]*, **Angelo Giuseppe Roncalli da Costa Oliveira**[1☯]

**1** Public Health Program, Universidade Federal do Rio Grande do Norte, Natal, Rio Grande do Norte, Brazil, **2** Medicine Department, Universidade Federal do Vale do São Francisco, Bahia, Petrolina, Brazil, **3** Physical Therapy Department, Universidade Federal do Rio Grande do Norte, Natal, Rio Grande do Norte, Brazil, **4** Health Sciences, Universidade Federal do Rio Grande do Norte, Natal, Rio Grande do Norte, Brazil, **5** Faculdade Uninassau, Natal, Rio Grande do Norte, Brazil, **6** Medicine Department, College of Health Sciences, Paraíba, Campina Grande, Brazil

☯ These authors contributed equally to this work.
* sanderson_assis@hotmail.com (SJCA); marcelloguedes21@hotmail.com (MBOGG)

## Abstract

### Background

SARS-CoV-2 causes the new coronavirus disease (COVID-19) and it is weakening all health systems. Therefore, the most vulnerable populations are exposed to harmful consequences, such as illness and death. Thus, this study aims to estimate the temporal effect of COVID-19 dissemination on social indicators of the Northeastern region of Brazil.

### Methods

An ecological time-series study was developed with the following: diagnosed cases of COVID-19 in the largest inland cities of Northeast Brazil, Human Development Index (HDI), poverty incidence, and Gini coefficient. Cities with high HDI, poverty rate, and Gini presented a larger number of patients.

### Results

It was observed by evaluating case trends that COVID-19 spreads unevenly in inland cities of the Northeastern region of Brazil.

### Conclusions

In this sense, we emphasize that regional health managers should support small cities with vulnerable population and social assistance.

**Data Availability Statement:** Datasets generated and/or analyzed during the current study are available in the Brazilian Institute of Geography and Statistics database. The dependent variable was the

cumulative cases of COVID-19 diagnosis in the twenty cities analyzed (https://covid.saude.gov.br/). The following independent variables were collected in the database of the Brazilian Institute of Geography and Statistics (https://ibge.gov.br/). The data is in the public domain and the reader can freely access it at any time.

**Funding:** The authors received no specific funding for this work.

**Competing interests:** The authors have declared that no competing interests exist.

## Introduction

At the end of 2019, a new SARS-CoV-2 coronavirus originated in Wuhan, China, was responsible for the severe COVID-19 disease in humans [1, 2]. Consequently, a pandemic was declared by the World Health Organization on March 11, 2020 [2].

Concerning the pandemic impact on populations' lives and health systems, an abrupt exposure of socioeconomic and environmental differences producing vulnerabilities could be observed [3], limiting individual and collective reaction power [4], and decreasing the available hospital capacity, especially in regions that already need such attention [5]. Previously, the focal outbreaks and slow progression of chronic health conditions did not allow society to note hidden vulnerabilities.

Since the beginning of the pandemic, the situation in Brazil has grown increasingly grim. The Brazilian socio-political reality may have contributed to the high numbers of rates and deaths by COVID-19. Chaired by a man with authoritarian leadership style, Brazil's governance during the pandemic has been described as tragic by several commentators since the president repeatedly resisted the recommendations made by scientific experts (i.e., social isolation and use of masks) [6, 7].

Brazil is a continental country with heterogeneous social scenarios divided into five regions. The northeastern region is the second largest region in Brazil and presents the highest percentage of black and brown races, together with northern region. Despite having great natural and cultural wealth, northeast region is characterized by high social inequality levels and concentration of income, reflecting lower educational levels, quality of life, and access to health and sanitation services [8].

In this context, Northeast Brazil becomes a perfect environment for observing the effects of inequitable access to formal education, healthy food, and health services and actions [9]. The mentioned region currently accounts for approximately 34% of the confirmed cases of COVID-19 in Brazil and 32% of the notified deaths [10]. More specifically, inland cities feel these conditions due to dependence on the network of goods and services of capital cities.

In developing countries, such as Brazil, even the wastewater surveillance system favors the study of COVID-19, while regions with low water supply quality and basic sanitation may become prone to further disease spread [11]. The low relative air humidity of the mentioned region reduces air filtration capacity by airway ciliary cells, making it difficult to remove particles, release mucus, and reduce innate body defense [12]. Moreover, the high range of daytime temperature in this region is associated with mortality due to pulmonary and cardiovascular diseases [13].

Since there is no way to mitigate socioeconomic conditions without public policies providing relief at this time of pandemic [14], the purpose of this study was to evaluate the progression of COVID-19 cases in the inland population of the Northeastern Brazilian region conditioned by socioeconomic indicators.

## Materials and methods

This ecological time-series study was performed with the largest inland cities of Northeast Brazil (population between 100 and 500 thousand inhabitants) outside metropolitan regions. Northeast Brazil has an area of 1558,000 km and a population of 56.56 million inhabitants. Historically, this region has the worst social and economic indicators in Brazil, and cities outside metropolitan regions present diverse biopsychosocial health conditions.

Among the twenty largest cities that are not capitals of federative units, 18 are not located in metropolitan areas: Alagoinhas, Arapiraca, Barreiras, Campina Grande, Caruaru, Caxias, Ilhéus, Itabuna, Jequié, Juazeiro, Juazeiro do Norte, Mossoró, Parnaíba, Petrolina, Porto

Seguro, Sobral, Teixeira de Freitas, and Vitória da Conquista. Only four cities have more than 300 thousand inhabitants and are considered medium-large.

All confirmed cases of COVID-19 present in the information system of the Unified Health System (created to monitor the pandemic) were assessed. Dependent variable was the cumulative cases of COVID-19 diagnosis in the twenty cities analyzed (https://covid.saude.gov.br/). The following independent variables collected in the database of the Brazilian Institute of Geography and Statistics (https://ibge.gov.br/) were analyzed: Human Development Index (HDI), Gini coefficient (Gini), and poverty rate [9]. Data were collected in June 2020 (22nd epidemiological week).

HDI is used to analyze the development of a given location and considers three main aspects of the population: income, education, and health. The higher the HDI value, the greater the development. Gini is used to measure social inequality through income concentration, and values range from 0 to 1 (values close to 1 indicate great inequality). Poverty index, on the other hand, is a measure of poverty in a given location, and higher values indicate the poorest locations.

Data were analyzed descriptively using mean and standard deviation. Independent variables were stratified following the United Nations [3] criteria and range of data variability. Thus, the following cut-off points were considered: inferior ($\leq$0.640), intermediate (0.641–0.690), and superior ($>$0.69) strata, for HDI; inferior ($\leq$0.45) and superior ($\geq$0.46), for Gini; and inferior ($\leq$42.96%), intermediate (42.97–52.17%), and superior ($\geq$52.18), for poverty rate.

We used chi-square linear trend tests to reduce bias regarding dependence between socioeconomic indicators in interpreting the effects on COVID-19 dissemination. This approach allowed verifying the effects of concentration of municipalities with high/low HDI and high/low Gini indexes.

Unpaired t-test was applied to estimate the effect of city size on number of cases. Medium- (100–300 thousand inhabitants) and medium-large (300–500 thousand inhabitants) cities were considered. Data analyses were performed using the SPSS, version 22 (IBM Corp., USA), and statistical significance was set at $p<0.05$.

Research ethics committee was unnecessary because this is a secondary data analysis.

## Results

In the 18 largest inland cities of Northeastern Brazil, 6,117 cases of COVID-19 were diagnosed during the 21st epidemiological week of 2020. The average number of cases per city was 305.85 ±318.52, varying from 32 to 1126 cases.

The included cities presented a mean Gini of 0.466±0.022 (minimum of 0.43 and maximum of 0.50) and mean poverty incidence of 48.23±7.23% (minimum of 33.69% and maximum of 60.44%). Mean HDI was 0.688±0.023, the lowest being 0.624 and the highest 0.721.

Case trends showed that COVID-19 spreads unevenly in inland cities of Northeastern Brazil. This distinction can be perceived by stratifying the evolution, according to HDI and social inequality. Fig 1 demonstrates that cities with higher HDI presented faster COVID-19 dissemination than those with lower HDI. Growth in the lower strata presented a lower gradient.

In those cities with Gini in the lower stratum, disease progression started later, generating a lower amount of diseased people (narrow apex) than those cities with coefficient of $>$0.46 (Fig 2). Regarding poverty rate, the lower the proportion of poverty people, the lower the slope of dissemination and symptomatic people (Fig 3).

To minimize interactions between HDI, Gini, and poverty, no association between high HDI and low Gini were observed ($x^2$ = 2.17; p = 0.14), as well as between Gini and poverty

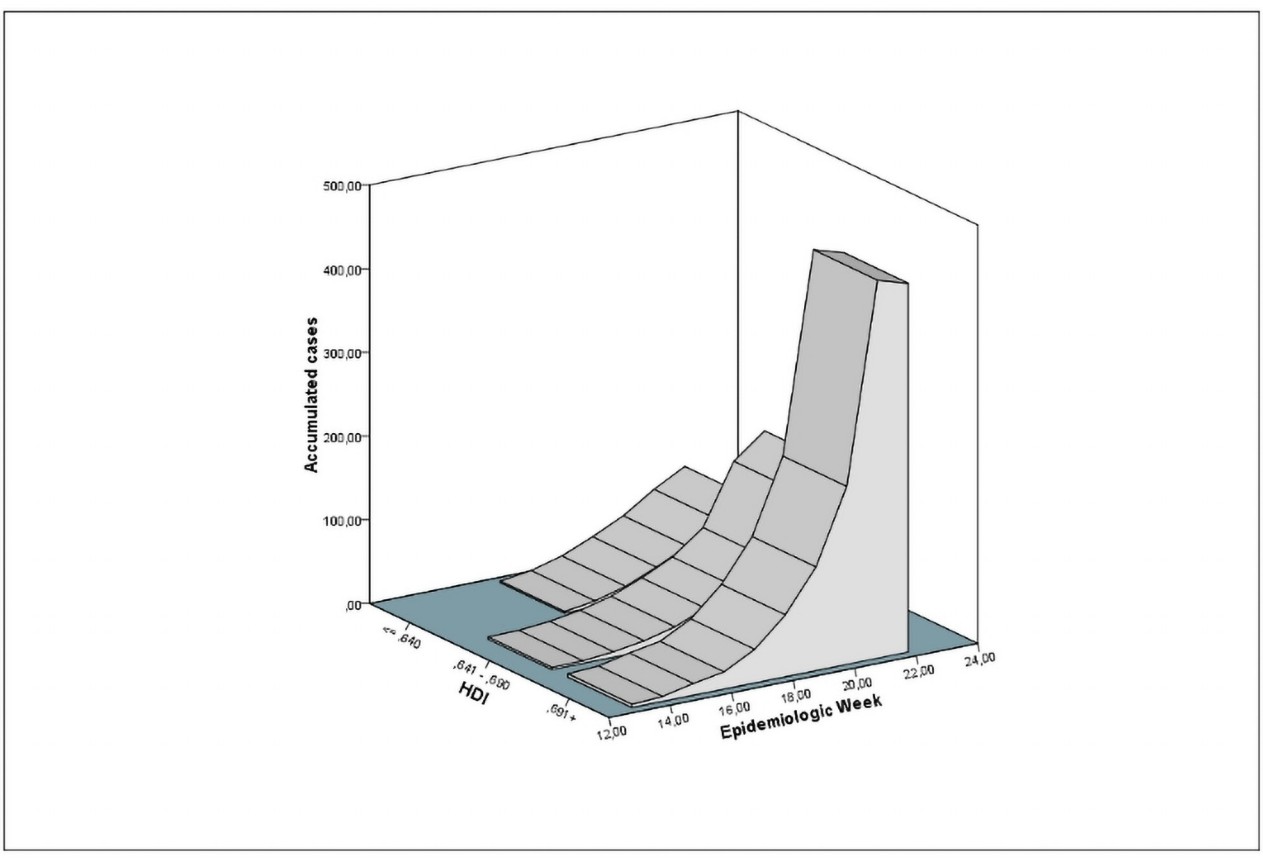

**Fig 1. Temporal distribution of COVID-19 cases stratified by the Human Development Index (HDI).**

incidence ($x^2$ = 3.53; p = 0.06), indicating the presence of cities with both social inequality and development (Table 1).

Regarding the effect of demographic size, no significant difference between medium- and medium-large cities (308.86 *vs.* 292.25; t = 0.82; p = 0.93) was observed.

To illustrate the scenario, the city of Sobral (located in the state of Ceará) presents the highest cumulative number of COVID-19 cases (538.92 diagnoses for every 100,000 inhabitants) and lies in the highest HDI and Gini strata and intermediate poverty incidence stratum. On the other hand, Itabuna (state of Bahia), with an accumulated incidence of 414.59 cases per 100,000 inhabitants, lies in the upper HDI and Gini strata, but in the lower poverty incidence stratum. Mossoró (state of Rio Grande do Norte) has the third higher incidence (accumulated incidence of 331.56 cases per 100,000 inhabitants) and is the only city with high HDI, Gini, and poverty incidence strata.

The lowest incidence rates were registered in Juazeiro (state of Bahia), 16.61 cases per 100,000 inhabitants, which falls into the intermediate HDI and poverty strata, and the upper Gini stratum. It is followed by Barreiras and Alagoinha (both in the state of Bahia), the first falling in the upper HDI stratum, lower poverty stratum, and intermediate Gini stratum; while the second differs only in the HDI.

## Discussion

All assessed indicators point to faster dissemination or diagnosis among cities in the upper HDI stratum. The following two hypotheses may explain this situation: cities with higher HDI

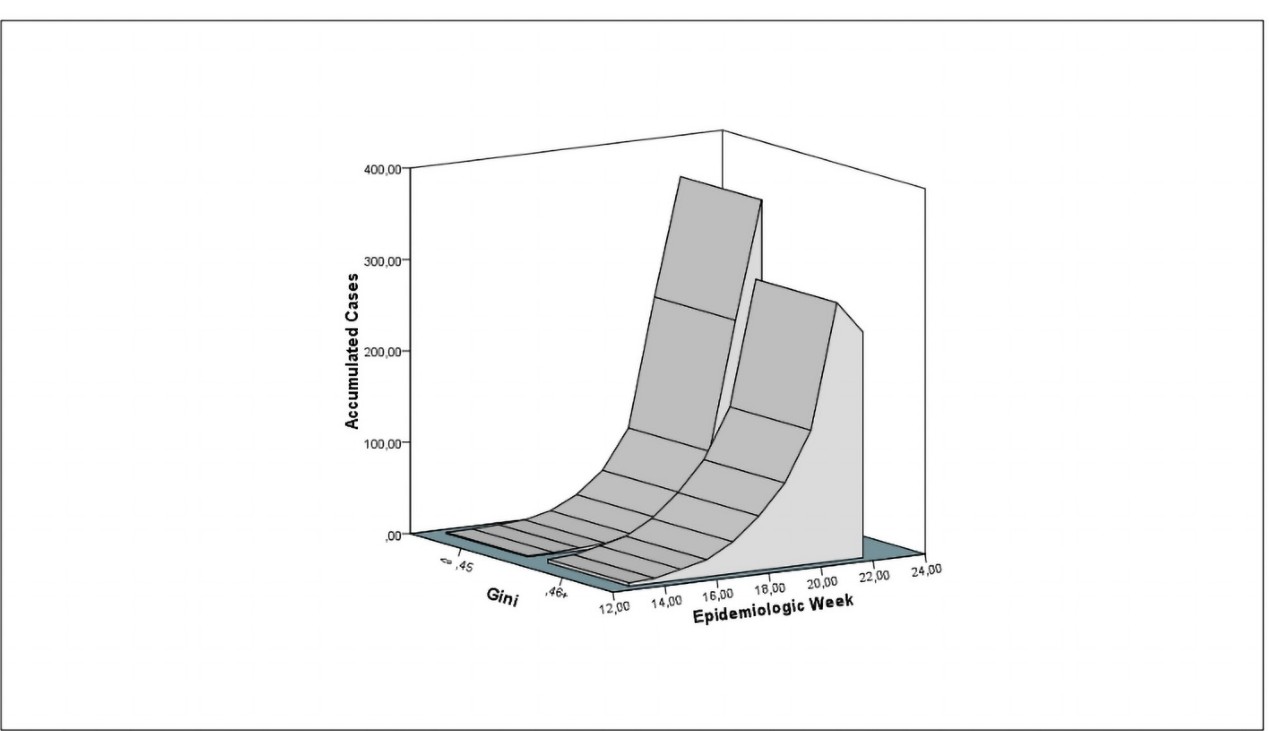

**Fig 2. Temporal distribution of COVID-19 cases stratified by the Gini coefficient.**

have higher Gross Domestic Product per person, allowing greater purchasing power, and generating greater mobility for travel and dynamism of goods and services outside the city, including epicenter areas [15]; on the other hand, these cities may have more access to rapid tests and confirmatory COVID-19 exams due to higher tax receipts. The opposite seems to be true for cities in the lower HDI strata. Both assumptions may be correct, but HDI influence is mitigated when other indicators are analyzed.

Cities with high HDI, but lower social inequality (e.g., Caruaru [Pernambuco] and Barreiras [Bahia]), highlights the lower rates of confirmed COVID-19 cases. The contrary is also true (e.g., Itabuna [Bahia], Sobral [Ceará], and Mossoró [Rio Gande do Norte]).

Not all cities fit perfectly in this logical model, mainly due to interferences of other factors not evaluated in the present study, such as temporary political and social measures and mandatory use of masks. Considering that Northeastern region concentrates 63.4% of small Brazilian cities, specific action is needed for those who are not always remembered in social policies [16].

The lower social inequality and poverty reflect the population's capacity to resist calamitous situations. Families with difficult access to food [17] and housing are vulnerable to social isolation practice. Also, health services for a correct diagnosis and treatment are sometimes restricted, mainly due to local infrastructure (e.g., favelas) [18].

The health service of medium- to large-sized inland cities of Northeast Brazil with higher socioeconomic indicators tends to overload faster due to spread of the virus. However, inequality and high poverty are ideal conditions for health service collapse [19]. Therefore, the care of the individual with COVID-19 in the Northeast region must be considered from the economic, social, and environmental points of view [20].

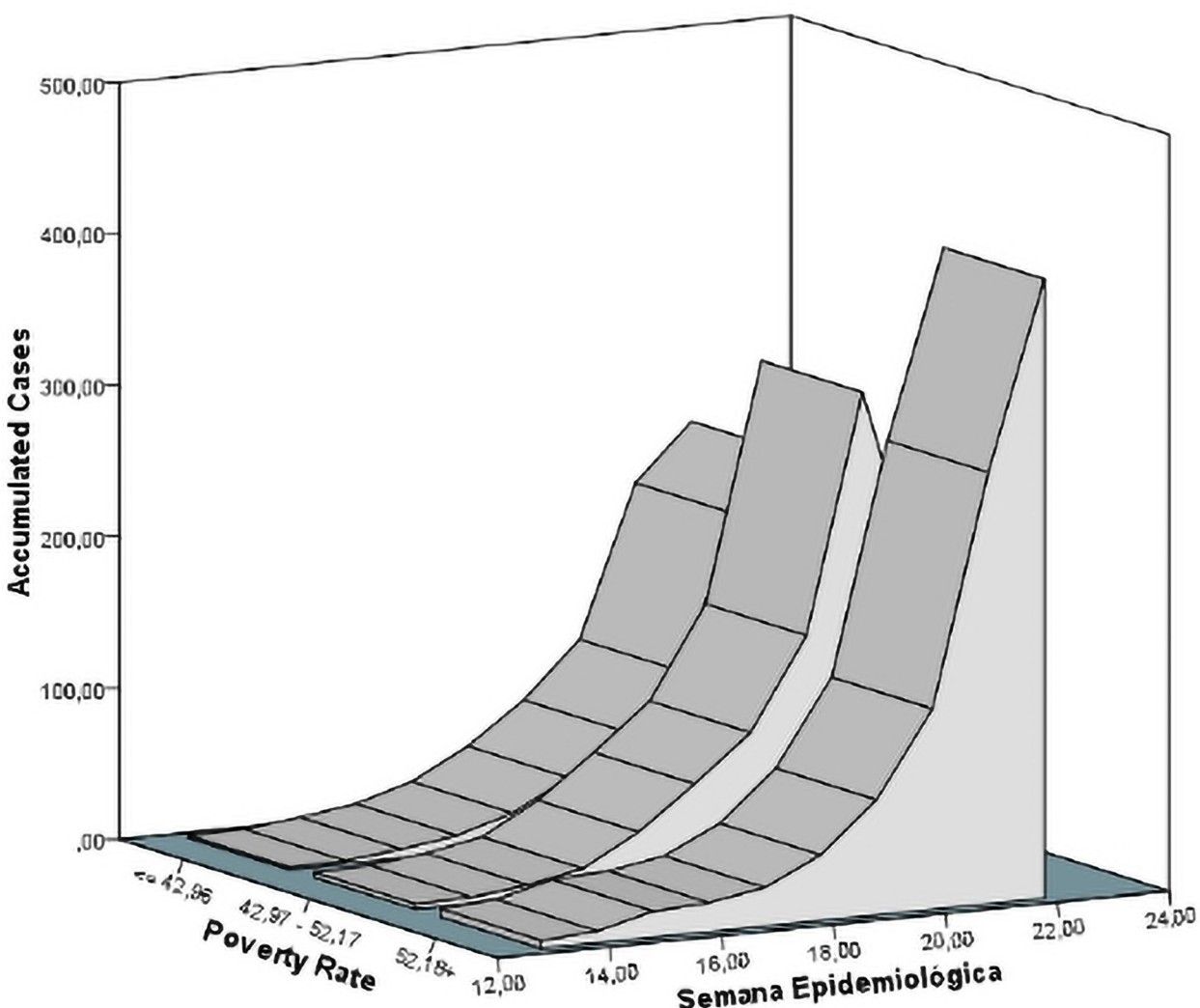

**Fig 3. Temporal distribution of COVID-19 cases stratified by the poverty incidence.**

**Table 1. Relationship between human development index, Gini coefficient, and poverty incidence in inland cities of Northeast Brazil, 2020.**

| Poverty incidence | | | Gini | | $x^2$ | p |
|---|---|---|---|---|---|---|
| | | | Inferior | Superior | | |
| Inferior | HDI | Intermediate | 1 (33.3%) | 2 (66.7%) | 0.99 | 0.31 |
| | | Superior | 0 (0.0%) | 3 (100.0%) | | |
| Intermediate | HDI | Intermediate | 0 (0.0%) | 4 (100.0%) | - | - |
| | | Superior | 0 (0.0%) | 2 (100.0%) | | |
| Superior | HDI | Inferior | 1 (100.0%) | 0 (0.0%) | 0.58 | 0.44 |
| | | Intermediate | 2 (66.7%) | 1 (33.3%) | | |
| | | Superior | 1 (50.0%) | 1 (50.0%) | | |
| Total | HDI | Inferior | 1 (100.0%) | 0 (100.0%) | 2.17 | 0.14 |
| | | Intermediate | 3 (30.0%) | 7 (70.0%) | | |
| | | Superior | 1 (14.3%) | 6 (85.7%) | | |

Adherence strategies to social isolation without adequate support to mitigate hunger, pre-existing health conditions, and habitation probably did not potentiate actions, such as increased availability of diagnostic tests and hospital beds. Actions to encourage the use and distribution of personal protective equipment for the population, popular education to prevent contamination, and implementation of public sanitization points are also suggested to combat virus dissemination and mitigate deaths resulted from this pandemic [21].

## Final considerations

COVID-19 reveals to be asymmetrical in medium- and large-sized inland cities of Northeast Brazil. Such differences contribute to socioeconomic characteristics of the cities, in which contamination is favored/identified faster in those with better HDI and mitigated in those with less social inequality and poverty. The combination of these factors leads to diverse scenarios for local and regional health managers.

## Supporting information

**S1 File.**
(XLS)

## Author Contributions

**Conceptualization:** Sanderson José Costa de Assis, Johnnatas Mikael Lopes, Bartolomeu Fagundes de Lima Filho, Geronimo José Bouzas Sanchis, Thais Sousa Rodrigues Guedes, Rafael Limeira Cavalcanti, Diego Neves Araujo, Antonio José Sarmento da Nóbrega, Marcello Barbosa Otoni Gonçalves Guedes.

**Data curation:** Sanderson José Costa de Assis, Johnnatas Mikael Lopes, Bartolomeu Fagundes de Lima Filho, Geronimo José Bouzas Sanchis, Thais Sousa Rodrigues Guedes, Rafael Limeira Cavalcanti, Diego Neves Araujo, Antonio José Sarmento da Nóbrega, Marcello Barbosa Otoni Gonçalves Guedes, Angelo Giuseppe Roncalli da Costa Oliveira.

**Formal analysis:** Johnnatas Mikael Lopes, Bartolomeu Fagundes de Lima Filho, Marcello Barbosa Otoni Gonçalves Guedes, Angelo Giuseppe Roncalli da Costa Oliveira.

**Investigation:** Johnnatas Mikael Lopes.

**Writing – original draft:** Sanderson José Costa de Assis, Johnnatas Mikael Lopes, Bartolomeu Fagundes de Lima Filho, Geronimo José Bouzas Sanchis, Thais Sousa Rodrigues Guedes, Rafael Limeira Cavalcanti, Diego Neves Araujo, Antonio José Sarmento da Nóbrega, Marcello Barbosa Otoni Gonçalves Guedes, Angelo Giuseppe Roncalli da Costa Oliveira.

**Writing – review & editing:** Sanderson José Costa de Assis, Johnnatas Mikael Lopes, Bartolomeu Fagundes de Lima Filho, Geronimo José Bouzas Sanchis, Thais Sousa Rodrigues Guedes, Rafael Limeira Cavalcanti, Diego Neves Araujo, Antonio José Sarmento da Nóbrega, Marcello Barbosa Otoni Gonçalves Guedes, Angelo Giuseppe Roncalli da Costa Oliveira.

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
