## [Decision Letter · Decision Letter 0]

17 Mar 2021

PONE-D-20-39302

DISSEMINATION OF COVID-19 IN INLAND CITIES OF NORTHEASTERN BRAZIL

PLOS ONE

Dear Dr.Sanderson Assis,

Thank you for submitting your manuscript to PLOS ONE. After careful consideration, we feel that it has merit but does not fully meet PLOS ONE’s publication criteria as it currently stands. Therefore, we invite you to submit a revised version of the manuscript that addresses the points raised during the review process.

Interesting because representing a secondary data analysis relating to smaller cities in the interior of NE Brazil and correlates the spread of the disease to socio-demographic indicators.

To make the article suitable for publication, it is necessary to perform the following actions:

1. To improve the description of the socio-economic and demographic context both of the examined region and of the whole of Brazil in relation to the Covid 19 pandemic

2. To add to the bibliography articles regarding socio-demographic studies carried out in Brazil and use of the indicators cited (HDI, GINI and poverty rate)

3. To improve the description of the data and the methods of analysis

4. To improve the number of observations.

5. To improve the discussion and the validity of the proposed model, include in the discussion the evaluation of the temporary political and social measures and the mandatory use of masks in the incidence of Covid cases 19 (see: IHME Covid-19 Projections: https://covid19.healthdata.org/global?view=total-deaths&tab=trend)

6. Line 97: correct 20202 with 2020

The decision is justified on PLOS ONE’s publication criteria 

Please submit your revised manuscript by 14, April 2021.  If you will need more time than this to complete your revisions, please reply to this message or contact the journal office at plosone@plos.org. Please include the following items when submitting your revised manuscript:

We look forward to receiving your revised manuscript.

Kind regards,

Filomena Pietrantonio

Academic Editor

PLOS ONE

Journal Requirements:

2. In your Methods section, please provide additional information about the methodology used, by listing the data sources used and describing the timeframe analysed and how data were extracted and analysed.

"No"

Additional Editor Comments:

The article is interesting because it represents a secondary data analysis relating to smaller cities in the interior of NE Brazil and correlates the spread of the disease to socio-demographic indicators.

To make the article suitable for publication, it is necessary to perform the following actions:

1. To improve the description of the socio-economic and demographic context both of the examined region and of the whole of Brazil in relation to the Covid 19 pandemic

2. To add to the bibliography articles regarding socio-demographic studies carried out in Brazil and use of the indicators cited (HDI, GINI and poverty rate)

3. To improve the description of the data and the methods of analysis

4. To improve the number of observations.

5. To improve the discussion and the validity of the proposed model, include in the discussion the evaluation of the temporary political and social measures and the mandatory use of masks in the incidence of Covid cases 19 (see: IHME Covid-19 Projections: https://covid19.healthdata.org/global?view=total-deaths&tab=trend)

6. Line 97: correct 20202 with 2020

Reviewers' comments:

Reviewer's Responses to Questions

**Comments to the Author**

1. Is the manuscript technically sound, and do the data support the conclusions?

Reviewer #1: Partly

Reviewer #2: Yes

2. Has the statistical analysis been performed appropriately and rigorously? 

Reviewer #1: No

Reviewer #2: Yes

3. Have the authors made all data underlying the findings in their manuscript fully available?

Reviewer #1: No

Reviewer #2: Yes

4. Is the manuscript presented in an intelligible fashion and written in standard English?

Reviewer #1: Yes

Reviewer #2: Yes

5. Review Comments to the Author

Reviewer #1: This is an important paper that presents findings relating to smaller cities in the interior of NE Brazil. The question it asks are important. However, the paper needs major revision in order to clarify and sustain its arguments. The soundness of the argument and the treatment of the data go hand in hand here. In brief, the paper attempts to derive statistically sound conclusions from a very small number of observations. This is possible, but complicated, and the authors need to do more to convince the reader that their approach is truly sound. For instance, the claim that the "independent" variables are independent of one another is perhaps technically true at .05 sig. but p-values should never be taken as the final arbiter or shortcut to this conclusion. Two substantive recommendations: 1) include access to the raw data and/or a table of values so that readers can assess these issues directly; 2) include scatter-plots rather than or in addition to the graphs in the paper (which are hard to read and need not be presented in three-dimensional representations). Including a more robust description of the data and the methods of analysis will greatly enhance this important paper. Finally, as for the independent variables, it seems that a basic demographic profile should also be included. As it stands, the authors have just used the 18 data points for Gini, HDI, and Poverty (54 data points total) as independent variables--all of which are easily obtained from official data sources. Given that the spread and morbidity of Covid-19 is clearly correlated with demographic factors, at the very least this should be controlled for. It is possible that all the 18 cities have the same demographic profile, but this should be tested and shown, not assumed.

Reviewer #2: In this manuscript, the authors explore the impact of the spread of the SARS-CoV-2 virus on cities in the northeast of Brazil. Using a mixed statistical set of methods including times-series analysis, the authors find that the distribution of the spread of COVID-19 cases is uneven and depends on a variety of mitigating factors such as inequality and vulnerability of the population. The authors conclude the public health officials in small cities need additional support to address the public health crisis.

This manuscript is timely and adds to our understanding about the current global pandemic in the developing world. The research methods are appropriate, and the conclusions are sound.

I recommend that the authors add some additional information about the socioeconomic and political situation in Brazil to add context in the introduction section. Specifically, I recommend the following minor revisions:

1. Define the geography of the Northeastern and situate it into the larger context of Brazil

2. Provide some basic demographic characteristics of this region in Brazil and how they compare to the country as a whole, including race/ethnicity, population, and income distribution

3. Reflect on the socio-political situation in Brazil and how it contributed to the pandemic worsening in Brazil. The COVID crisis in Brazil has been exacerbated by the political crisis in Brazil and lack of political leadership of science.

4. Integrate some scholarly literature that adds to the understanding of modern Brazil today. There is a vast literature on Brazil's development as a new global power, including "Brazil: A Biography" by Schwarcz and Starling; "Brazil: The Troubled Rise of a Global Power" by Michael Reid; "Brazil on the Rise" by Larry Rohter, among many others. A brief reflection on the context of Brazil in the world will improve the readability of the article for an international audience.

I recommend a minor revision and look forward to reading the published article. Thank you.

6. PLOS authors have the option to publish the peer review history of their article (what does this mean?). If published, this will include your full peer review and any attached files.

Reviewer #1: No

Reviewer #2: No

---

## [Author Response · Author response to Decision Letter 0]

10 Apr 2021

Filomena Pietrantonio

Editor-in-Chief

Plos One

 April 06, 2021.

Dear Filomena Pietrantonio,

 Thank you for your email with the reviewers’ comments. We have reviewed the comments and edited the manuscript accordingly. Please, find attached our point-by-point response to the reviewers. All authors have read this protocol and agreed with Plos One policy. We hope the revised manuscript is now suitable for publication. 

Sincerely. Sanderson José Costa de Assis.

Reviewer Comments:

Additional Editor Comments: The article is interesting because it represents a secondary data analysis relating to smaller cities in the interior of NE Brazil and correlates the spread of the disease to socio-demographic indicators.

To make the article suitable for publication, it is necessary to perform the following actions:

1. To improve the description of the socio-economic and demographic context both of the examined region and of the whole of Brazil in relation to the Covid 19 pandemic.

Response: Thank you for your comments. The following sentences were added:

“Brazil is a continental country with heterogeneous social scenarios divided into five regions. The northeastern region is the second largest region in Brazil and presents the highest percentage of black and brown races, together with northern region. Despite having great natural and cultural wealth, northeast region is characterized by high social inequality levels and concentration of income, reflecting lower educational levels, quality of life, and access to health and sanitation services.”

“In this context, Northeast Brazil becomes a perfect environment for observing the effects of inequitable access to formal education, healthy food, and health services and actions.”

2. To add to the bibliography articles regarding socio-demographic studies carried out in Brazil and use of the indicators cited (HDI, GINI and poverty rate).

Response: Thank you for your comments. The following sentence was added:

“HDI is used to analyze the development of a given location and considers three main aspects of the population: income, education, and health. The higher the HDI value, the greater the development. Gini is used to measure social inequality through income concentration, and values range from 0 to 1 (values close to 1 indicate great inequality). Poverty index, on the other hand, is a measure of poverty in a given location, and higher values indicate the poorest locations.”

3. To improve the description of the data and the methods of analysis

Response: Thank you for your comments. The sentence was rewritten as follows:

“We used chi-square linear trend tests to reduce bias regarding dependence between socioeconomic indicators in interpreting the effects on COVID-19 dissemination. This approach allowed verifying the effects of concentration of municipalities with high/low HDI and high/low Gini indexes.

Unpaired t-test was applied to estimate the effect of city size on number of cases. Medium- (100-300 thousand inhabitants) and medium-large (300-500 thousand inhabitants) cities were considered. Data analyses were performed using the SPSS, version 22 (IBM Corp., EUA), and statistical significance was set at p<0.05.”

4. To improve the number of observations.

Response: Thank you for your comments. The sentence was rewritten:

“Among the twenty largest cities that are not capitals of federative units, 18 are not located in metropolitan areas”

5. To improve the discussion and the validity of the proposed model, include in the discussion the evaluation of the temporary political and social measures and the mandatory use of masks in the incidence of Covid cases 19 (see: IHME Covid-19 Projections: https://covid19.healthdata.org/global?view=total-deaths&tab=trend).

Response: Thank you for your comments. The sentence was rewritten:

“Not all cities fit perfectly in this logical model, mainly due to interferences of other factors not evaluated in the present study, such as temporary political and social measures and mandatory use of masks. Considering that Northeastern region concentrates 63.4% of small Brazilian cities, specific action is needed for those who are not always remembered in social policies.”

6. Line 97: correct 20202 with 2020

Response: Thank you for your comments. The sentence was rewritten:

“Consequently, a pandemic was declared by the World Health Organization on March 11, 2020.”

Reviewer #1: This is an important paper that presents findings relating to smaller cities in the interior of NE Brazil. The question it asks are important. However, the paper needs major revision in order to clarify and sustain its arguments. The soundness of the argument and the treatment of the data go hand in hand here. In brief, the paper attempts to derive statistically sound conclusions from a very small number of observations. This is possible, but complicated, and the authors need to do more to convince the reader that their approach is truly sound. For instance, the claim that the "independent" variables are independent of one another is perhaps technically true at .05 sig. but p-values should never be taken as the final arbiter or shortcut to this conclusion. Two substantive recommendations:

1. include access to the raw data and/or a table of values so that readers can assess these issues directly.

Response: Thanks for your comment. The following sentence has been modified to include locations where the data can be accessed:

“All confirmed cases of COVID-19 present in the information system of the Unified Health System (created to monitor the pandemic) were assessed. Dependent variable was the cumulative cases of COVID-19 diagnosis in the twenty cities analyzed (https://covid.saude.gov.br/). The following independent variables collected in the database of the Brazilian Institute of Geography and Statistics (https://ibge.gov.br/) were analyzed: Human Development Index (HDI), Gini coefficient (Gini), and poverty rate. Data were collected in June 2020 (22nd epidemiological week).”

2. include scatter-plots rather than or in addition to the graphs in the paper (which are hard to read and need not be presented in three-dimensional representations).

Response: 

We thank the reviewer for the observation. However, we understand that a three-dimensional column chart represents the gradient aspect in the HDI strata very well. The scatter-plot with three variables, two of which are qualitative, does not convey the information well, as you can see below.

3. Including a more robust description of the data and the methods of analysis will greatly enhance this important paper.

Response: 

We appreciate the comments of the reviewer.

We added information on the methodology and results that expand data description and spatial context of the study.

4. Finally, as for the independent variables, it seems that a basic demographic profile should also be included. As it stands, the authors have just used the 18 data points for Gini, HDI, and Poverty (54 data points total) as independent variables--all of which are easily obtained from official data sources. Given that the spread and morbidity of Covid-19 is clearly correlated with demographic factors, at the very least this should be controlled for. It is possible that all the 18 cities have the same demographic profile, but this should be tested and shown, not assumed.

Response: We are grateful for the pertinent observation! An inferential analysis was now added in the results section to estimate differences in number of cases according to the size of the city. However, no significant differences were found.

Reviewer #2: In this manuscript, the authors explore the impact of the spread of the SARS-CoV-2 virus on cities in the northeast of Brazil. Using a mixed statistical set of methods including times-series analysis, the authors find that the distribution of the spread of COVID-19 cases is uneven and depends on a variety of mitigating factors such as inequality and vulnerability of the population. The authors conclude the public health officials in small cities need additional support to address the public health crisis.

This manuscript is timely and adds to our understanding about the current global pandemic in the developing world. The research methods are appropriate, and the conclusions are sound.

I recommend that the authors add some additional information about the socioeconomic and political situation in Brazil to add context in the introduction section. Specifically, I recommend the following minor revisions:

1. Define the geography of the Northeastern and situate it into the larger context of Brazil

Response: Thank you for your comments. The sentence was added as follows:

“Brazil is a continental country with heterogeneous social scenarios divided into five regions. The northeastern region is the second largest region in Brazil and presents the highest percentage of black and brown races, together with northern region. Despite having great natural and cultural wealth, northeast region is characterized by high social inequality levels and concentration of income, reflecting lower educational levels, quality of life, and access to health and sanitation services.”

2. Provide some basic demographic characteristics of this region in Brazil and how they compare to the country as a whole, including race/ethnicity, population, and income distribution

Response: Thank you for your comments. The sentence was rewritten:

“The northeastern region is the second largest region in Brazil and presents the highest percentage of black and brown races, together with northern region. Despite having great natural and cultural wealth, northeast region is characterized by high social inequality levels and concentration of income, reflecting lower educational levels, quality of life, and access to health and sanitation services.” 

In this context, Northeast Brazil becomes a perfect environment for observing the effects of inequitable access to formal education, healthy food, and health services and actions.”

3. Reflect on the socio-political situation in Brazil and how it contributed to the pandemic worsening in Brazil. The COVID crisis in Brazil has been exacerbated by the political crisis in Brazil and lack of political leadership of science.

Response: Thank you for your comments. The sentence was added:

“Since the beginning of the pandemic, the situation in Brazil has grown increasingly grim. The Brazilian socio-political reality may have contributed to the high numbers of rates and deaths by COVID-19. Chaired by a man with authoritarian leadership style, Brazil’s governance during the pandemic has been described as tragic by several commentators since the president repeatedly resisted the recommendations made by scientific experts (i.e., social isolation and use of masks)”

4. Integrate some scholarly literature that adds to the understanding of modern Brazil today. There is a vast literature on Brazil's development as a new global power, including "Brazil: A Biography" by Schwarcz and Starling; "Brazil: The Troubled Rise of a Global Power" by Michael Reid; "Brazil on the Rise" by Larry Rohter, among many others. A brief reflection on the context of Brazil in the world will improve the readability of the article for an international audience.

Response: Thank you for your comments. The sentence was added:

“Since the beginning of the pandemic, the situation in Brazil has grown increasingly grim. The Brazilian socio-political reality may have contributed to the high numbers of rates and deaths by COVID-19. Chaired by a man with authoritarian leadership style, Brazil’s governance during the pandemic has been described as tragic by several commentators since the president repeatedly resisted the recommendations made by scientific experts (i.e., social isolation and use of masks)”

All changes made are highlighted in the manuscript.

Thank you for your comment. The manuscript has been revised accordingly. 

Sincerely,

Sanderson José Costa de Assis. Federal University of Rio Grande do Norte. Corresponding author. Natal, Rio Grande do Norte, Brazil.

Mobile: +5584996219425

e-mail: sanderson_assis@hotmail.com

---

## [Decision Letter · Decision Letter 1]

31 May 2021

DISSEMINATION OF COVID-19 IN INLAND CITIES OF NORTHEASTERN BRAZIL

PONE-D-20-39302R1

Dear Dr. Sanderson José Costa de Assis,

We’re pleased to inform you that your manuscript has been judged scientifically suitable for publication and will be formally accepted for publication once it meets all outstanding technical requirements.

Kind regards,

Filomena Pietrantonio

Academic Editor

PLOS ONE

Additional Editor Comments :

The revised paper has responded to all reviewers' requests and therefore can be accepted for publication.

Reviewers' comments:

Reviewer's Responses to Questions

**Comments to the Author**

The revised paper has responded to all reviewers' requests and therefore can be accepted for publication.

Reviewer #1: All comments have been addressed

Reviewer #2: All comments have been addressed

2. Is the manuscript technically sound, and do the data support the conclusions?

Reviewer #1: Yes

Reviewer #2: Yes

3. Has the statistical analysis been performed appropriately and rigorously? 

Reviewer #1: Yes

Reviewer #2: Yes

4. Have the authors made all data underlying the findings in their manuscript fully available?

Reviewer #1: Yes

Reviewer #2: Yes

5. Is the manuscript presented in an intelligible fashion and written in standard English?

Reviewer #1: Yes

Reviewer #2: Yes

6. Review Comments to the Author

Reviewer #1: The revised paper is strong and convincing. While one may always quibble about data display (I follow Tufte in avoiding three dimensional mass plots when possible), the research question, data, and discussion are all presented clearly. In addition, this is very important and timely research with possible public health ramifications.

Reviewer #2: All comments have been addressed. The authors have incorporated relevant literature and further contextualized the case of the Brazil for an international audience. Thank you.

7. PLOS authors have the option to publish the peer review history of their article (what does this mean?). If published, this will include your full peer review and any attached files.

Reviewer #1: No

Reviewer #2: No

---

## [Editor Report · Acceptance letter]

29 Jun 2021

PONE-D-20-39302R1 

Dissemination of COVID-19 in inland cities of Northeastern Brazil 

Dear Dr. Costa de Assis:

I'm pleased to inform you that your manuscript has been deemed suitable for publication in PLOS ONE. Congratulations! Your manuscript is now with our production department. 

Kind regards, 

on behalf of

Dr. Filomena Pietrantonio 

Academic Editor

PLOS ONE